# Combination Therapy for HIV-Associated Cryptococcal Meningitis—A Success Story

**DOI:** 10.3390/jof7121098

**Published:** 2021-12-20

**Authors:** William J. Hurt, Thomas S. Harrison, Síle F. Molloy, Tihana A. Bicanic

**Affiliations:** 1Institute of Infection & Immunity, St George’s University London, London SW17 0RE, UK; tharriso@sgul.ac.uk (T.S.H.); smolloy@sgul.ac.uk (S.F.M.); tbicanic@sgul.ac.uk (T.A.B.); 2Clinical Academic Group in Infection & Immunity, St George’s University Hospitals NHS Trust, London SW17 0QT, UK; 3The MRC Centre of Medical Mycology, University of Exeter, Stocker Road, Exeter EX4 4QD, UK

**Keywords:** cryptococcal meningitis, cryptococcus, antifungal treatment, HIV

## Abstract

Cryptococcal meningitis is the leading cause of adult meningitis in patients with HIV, and accounts for 15% of all HIV-related deaths in sub-Saharan Africa. The mainstay of management is effective antifungal therapy, despite a limited arsenal of antifungal drugs, significant progress has been made developing effective treatment strategies by using combination regimens. The introduction of fluconazole as a safe and effective step-down therapy allowed for shorter courses of more fungicidal agents to be given as induction therapy, with higher doses achieving more rapid CSF sterilisation and improved treatment outcomes. The development of early fungicidal activity (EFA), an easily measured surrogate of treatment efficacy, has enabled rapid identification of effective combinations through dose ranging phase II studies, allowing further evaluation of clinical benefit in targeted phase III studies. Recent clinical trials have shown that shorter course induction regimens using one week of amphotericin paired with flucytosine are non-inferior to traditional two-week induction regimens and that the combination of fluconazole and flucytosine offers a viable treatment alternative when amphotericin is unavailable. Access to drugs in many low and middle-income settings remains challenging but is improving, and novel strategies based on single high dose liposomal amphotericin B promise further reduction in treatment complications and toxicities. This review aims to summarise the key findings of the principal clinical trials that have led to the success story of combination therapy thus far.

## 1. Introduction

Cryptococcal meningitis (CM) is a severe infection caused by the environmental yeasts *Cryptococcus neoformans* and *Cryptococcus gattii*. CM is primarily seen in patients with impaired cell-mediated immunity, most commonly in the context of advanced HIV (CD4 count < 200) [1]. It remains the leading cause of adult meningitis in sub-Saharan Africa where it accounts for 15% of all HIV-related mortality, and is estimated to cause 181,800 deaths annually [1].

Antifungal therapy is the mainstay of management, with a mortality rate of 100% in untreated disease [1,2]. There are currently only three antifungal classes licensed for use in CM; the polyenes, azoles, and pyrimidine analogues. While the development of novel therapeutics will hopefully strengthen the armamentarium of treatment options against CM [3], considerable success has been achieved by optimising combinations of existing drugs. Optimised combination therapy can achieve rapid clearance of infection from the cerebrospinal fluid (CSF), reduce relapse rates, minimise emergent drug resistance, allow a reduction in the duration and hence toxicity of component agents, and ultimately improve patient outcomes when compared to monotherapy. This review aims to summarise the key findings of the principal clinical studies that have led to the success story of combination therapy thus far.

## 2. Available Antifungal Agents

Amphotericin deoxycholate (AmBd) is a polyene which disrupts fungal cell integrity by binding to ergosterol in the cell membrane. With excellent fungicidal activity [4], AmBd has formed the backbone of CM treatment for over 50 years [5] and is still recommended as part of first line therapy [6,7]. Unfortunately its fungicidal properties are tempered by adverse effects, which include anaemia, hypokalaemia, hypomagnesaemia, nephrotoxicity, and infusion site thrombophlebitis [8]. These side effects limit tolerability and contribute to adverse patient outcomes, especially when given for prolonged periods [9]. The requirement for frequent laboratory monitoring and intravenous (IV) administration can prohibit its use in some resource poor settings [10], and while a liposomal formulation (L-AmB) is available and has a more favourable side effect profile [11], it is significantly more costly [12].

Flucytosine (5-FC) is a pyrimidine analogue whose active metabolite 5-Flourouracil inhibits fungal RNA and DNA synthesis [13]. While active against *Cryptococcus*, given as a single agent it quickly induces stable and highly resistant mutants, precluding its use as monotherapy [14]. The most frequently encountered toxicity is bone marrow suppression; although, this is dose dependent and can be largely avoided with appropriate dosing [9]. While available in oral and IV formulations, 5-FC must currently be administered four times daily. Cost [15], and access has been an issue [12], but following the recent ACTA trial [16], advocacy efforts, and a programme launched by Unitaid [17], generic 5-FC at reduced cost is increasingly available and is being used in routine care in South Africa.

Fluconazole (FCZ) is a triazole that acts at the fungal cytochrome P450 to stop the conversion of lanosterol into ergosterol, thereby disrupting the fungal cell membrane [18]. Orally administered, widely available and with a favourable side effect profile, it is still, at the time of writing, often used as monotherapy in settings where AmBd and 5-FC are impractical, unaffordable, or unavailable [15,19]. Despite being slower to clear cryptococcus from the CSF, FCZ was initially proposed as an effective treatment alternative to AmBd-based regimes for patients at low risk of treatment failure [20]. Subsequent trials in people living with HIV, however, have demonstrated that FCZ monotherapy is associated with an unacceptably high 10-week mortality of 50–60%, despite the use of higher doses of FCZ (800–1200 mg vs. 200–400 mg) and concurrent improvements in access to antiretroviral therapy (ART) [19,21,22,23]. Persistence of *Cryptococcus* in the presence of subtherapeutic doses of FCZ encourages emergence of drug resistance, resulting in disease relapse [24] and contributes to FCZ resistance at a population level [25]. *Cryptococcus neoformans* exhibits intrinsic heteroresistance to FCZ mediated most commonly through aneuploidy, the most frequent being disomy of chromosome 1, where the drug target gene ERG11 and the efflux pump gene AFR1 are located. Undetectable by conventional resistance testing, Stone et al. [26] found that heteroresistant subpopulations were selected for in the presence of FCZ monotherapy, resulting in clinical relapse with resistance. Combination therapy of FCZ and 5-FC effectively suppresses this phenomenon, improving mycological clearance [26].

Other members of the triazole drug class have activity against *Cryptococcus* spp., [7,27,28] as shown by Lose et al. who found that voriconazole given with AmBd had an equivalent fungicidal activity to the combination of FCZ + AmBd [28], but are limited by poor CSF penetration (itraconazole) [29], frequent drug–drug interactions, especially with commonly prescribed ART (voriconazole and Itraconazole), [30] as well as cost and availability (posaconazole, and voriconazole).

## 3. Other Considerations in Successful Cryptococcal Management

The successful management of cryptococcal meningitis relies on more than just timely initiation of appropriate antifungal therapy. Raised intracranial pressure (ICP) is a common complication, occurring in 60–80% of patients at presentation [31,32] and can occur without associated symptoms [32,33]. Reducing raised ICP by performing therapeutic lumbar punctures has been shown to convey a survival benefit, regardless of the baseline opening pressure [33], and proactive ICP management through serial lumbar punctures is advised in clinical guidelines [6,7].

The timing of ART initiation is also an important consideration. While rapid initiation of ART reduces AIDS progression and death [34], these benefits must be weighed against the risk of immune reconstitution inflammatory syndrome (IRIS), a paradoxical reaction caused by immune recovery. When considering patients with a range of opportunistic infections, starting ART within 2 weeks was not associated with adverse events from IRIS and resulted in fewer overall deaths from AIDS progression when compared with starting ART at 6–12 weeks [35]. Makadzange et al. found that in patients with CM treated with FCZ monotherapy ART initiation at 72 hours resulted in increased mortality when compared to delayed initiation at 10 weeks [36] and Bisson et al. found that in patients treated with AmBd monotherapy while initiation of ART at 7 days improved CSF fungal clearance compared to initiation at 28 days, it also resulted in a higher risk of IRIS [37]. The COAT trial by Boulware et al. was the largest of this kind (n = 177) and found that delaying ART for 5 weeks after the initiation of antifungal therapy (AmBd + FCZ) significantly improved survival compared to starting ART at 1–2 weeks [38].

## 4. Early Combination Studies

Consistent in vitro and animal model studies have suggested that AmBd plus 5-FC, and FCZ plus 5-FC are additive or synergistic [39]. In some fungal infections the combination of polyenes and azoles has been reported to be antagonistic, as both classes affect the integrity of the fungal cell membrane by either depleting (azoles) or binding (polyenes) to ergosterol [18,40]; however, this does not appear to be the case in *Cryptococcus* [39,41,42]. While a detailed review of in vitro and animal studies is beyond the scope of this review, it is worth noting that the generalisability of in vitro studies to the clinical setting is limited. Careful PK/PD animal model studies, and ultimately, clinical trials are essential to establish the efficacy of drug combinations and to achieve optimal dosing in the setting of a compromised host [39].

The first randomised control trial of combination therapy for CM was performed in 1979, in the pre-HIV era and before the availability of FCZ. It compared 10-weeks of AmBd monotherapy given at a dose 0.4 mg/kg/day, to 6-weeks of AmBd (0.4 mg/kg/day) with 5-FC dosed at 150 mg/kg/day. Clinical improvement or cure at 10 weeks was 68% in the 6-week combination arm vs. 47% in the 10-week monotherapy arm. Combination therapy resulted in more rapid CSF sterilisation, fewer disease relapses, and less nephrotoxicity [43]. Further reduction in course length using an AmBd 0.3 mg/Kg/day + 5-FC 150 mg/kg/day regimen given for four weeks resulted in higher relapse rates compared to a 6-week treatment course (27% vs. 16%), despite similar initial cure rates [44]. These early studies established six weeks of AmBd + 5-FC as the gold standard in (non-HIV) CM treatment [45] and highlight the difficult balance between toxicity and risk of relapse when considering shorter course therapy in non-HIV-associated CM.

Unfortunately, the success of the AmBd 0.4 mg/kg/day + 5-FC 150 mg/kg/day regimen did not translate to those with advanced HIV-associated CM, albeit in the era prior to access to antiretroviral therapy [46,47,48]. 5-FC at a dose of 150 mg/kg was poorly tolerated, with discontinuation in up to 50% of patients due to cytopenias [47]. The triallists from the US Mycoses Study group adapted by (a) giving higher doses of AmBd but for a shorter 2-week period, following evidence from a small RCT where 6/6 patients treated with a higher dose of 0.7 mg/kg/d of AmBd with 5-FC survived at 10 weeks [49]; and (b) reducing the dose of 5-FC, in accordance with evidence which showed that 5-FC toxicity was more commonly seen when serum concentrations exceeded 100 ug/mL [9,44]. This coincided with the advent of the azoles as a new drug class which offered the possibility of oral stepdown therapy and allowed treatment to be split into an induction phase, which prioritises rapid fungal clearance, followed by a consolidation phase.

This trial was designed as a two-part RCT and was performed in a cohort of 381 patients with HIV-associated CM [50]. The first phase compared an induction regimen of AmBd at the higher dose of 0.7 mg/kg/day either with or without the addition of 5-FC dosed at 100 mg/kg/day (25 mg/kg in four divided doses) for 2 weeks. CSF sterility at two weeks was achieved in a greater proportion of patients receiving 5-FC combination therapy compared to AmBd monotherapy (60% vs. 51%) and use of 5-FC at induction and FCZ consolidation therapy were each independently associated with CSF sterility at ten weeks. Despite showing an association between CSF sterility and survival, there was no difference in clinical outcomes between treatment arms at two weeks, although overall mortally appeared improved when compared to historical dosing regimens (4–10 weeks’ AmBd 0.3–0.5 mg/Kg/day + 5-FC 150 mg/kg/day) [20,43,50]. It was also shown that shorter course induction therapy combined with the lower dosing of 5-FC resulted in reduced toxicity in the form of nephrotoxicity and cytopenias.

## 5. Paradigm Shift

The efficacy of longer-term FCZ in preventing relapse in HIV-CM patients was confirmed by Powderly et al. in a RCT of 213 patients prescribed either once weekly AmBd (1 mg/kg/day) or FCZ 200 mg daily for 12 months [51]. Two percent of patients in the FCZ arm had relapsed at 1 year compared to 14% in the AmBd arm. These findings, taken together with those of Van der Horst et al. [50], enabled a more nuanced approach when designing CM regimens, with a split into induction, consolidation, and maintenance phases. While a rapidly fungicidal agent or combination of agents is prioritised during the induction phase, a well-tolerated oral drug is preferable for longer-term suppression of relapse.

The findings of Van der Vorst et al. provided evidence that higher dosing of AmBd was effective and suggested that the inclusion of 5-FC increased fungicidal activity [50]; however, this trial was underpowered to detect a mortality difference at two weeks. The use of binary clinical and mycological endpoints such as 2- and 10-week mortality and CSF sterility required large numbers to achieve sufficient power: thus in order to explore differences between novel treatment approaches and prioritise those to take forward into large, costly phase III trials, a reliable surrogate marker was needed that allowed for smaller dose ranging studies to reach statistical significance [52]. The solution lay in the development of a quantitative measure of fungal clearance that could be calculated during therapy. Lumbar punctures are required for the diagnosis and ongoing pressure management of patients with CM. This is advantageous as it allows serial sampling at the affected site (central nervous system) over the course of induction treatment. Termed early fungicidal activity (EFA), this development was based on studies of early bactericidal activity (EBA) derived from serial quantitative sputum cultures developed in tuberculosis trials [53]. Serial CSF samples are diluted and cultured on solid media to calculate the colony forming units per ml of CSF, allowing the rate of clearance (the decrease in cryptococcal colony forming units in CSF) to be plotted against time for each patient. The mean rate of clearance for patients in the same treatment group can be calculated using linear regression or mixed effects models and is termed the EFA for that drug or combination regimen [4,16]. As a continuous measure, comparisons of EFA in relatively small numbers of patients (15–20 per arm) were able to demonstrate significant differences between arms [4,22,28,54,55]. EFA was subsequently shown to be independently associated with reduced mortality in large African combined clinical trial cohorts [52,56,57]

The development of EFA as a statistically powerful phase II surrogate endpoint for induction regimen efficacy (although not capturing issues of toxicity), and the success of FCZ consolidation therapy in preventing disease relapse represented a step change on the path to CM treatment optimisation.

## 6. Early Fungicidal Activity Studies of Combination Therapy for Induction Regimens

### 6.1. Amphotericin and Flucytosine

Brouwer et al.’s 2004 RCT was the first to use EFA as a primary outcome measure and demonstrated statistically significant differences between regimens using only 16 patients per arm [4]. The study compared AmBd 0.7 mg/kg/day alone; AmBd 0.7 mg/kg/d + 5-FC 100 mg/kg/day; AmBd 0.7 mg/kg/day + FCZ 400 mg/day; and AmBd 0.7 mg/kg/day + 5-FC 100 mg/kg/day + FCZ 400 mg/day (triple therapy). AmBd + 5-FC had the highest EFA of −0.54 log CFU/mL/day, compared to −0.31 for AmBd monotherapy and −0.39 and −0.38 for AmBd + 5-FC and AmBd + 5-FC + FCZ combinations, respectively.

Consistent with the concentration-dependent activity of AmBd, and following on the success of increasing AmBd dosing [6,50], Bicanic et al. compared the safety and efficacy of AmBd dosed at 1 mg/kg/day with 5-FC 100 mg/kg/day vs. the standard AmBd 0.7 mg/kg/day with 5-FC 100 mg/kg/day in 64 HIV positive patients [55]. AmBd dosed at 1.0 mg/kg/day with 5-FC had an EFA of −0.56 log CFU/mL/day compared to −0.45 log CFU/mL/day in the 0.7 mg/kg/day group, a difference which was significant in unadjusted and adjusted analyses. Anaemia did occur more commonly in the group treated with 1.0 mg/kg/day of AmBd with an average decrease in haemoglobin of 2.5 g/dL over the treatment course compared to 16 g/dL in the 0.7 mg/kg/day group; although, all episodes resolved two weeks post induction treatment and toxicity was rare in the first week of treatment.

The real-world benefits of AmBd + 5-FC combination induction regimens were highlighted by Dromer et al. [58,59] in a prospective observational study of 173 CM patients in France (144 HIV positive). A total of 49% received the recommended first line therapy of combination AmBd + 5-FC, 21% received AmBd monotherapy, 12% FCZ monotherapy, and 18% alternative combinations. Despite a higher proportion of severe disease in those treated with AMB + 5-FC, 2-week mycological failure was significantly reduced compared to any other regimen (23% vs. 47%, *p* < 0.001). The inclusion of 5-FC in any regimen was also independently associated with CSF sterilisation at two weeks [59]. These findings are notable as they did not exclude patients with severe disease as in some prior RCTs [50], reflected by the high proportion (40%) of patients with abnormal mental status at baseline.

With several phase II studies suggesting the superior EFA of AmBd 1.0 mg/kg/day+ 5-FC 100 mg/kg/day combination induction regimens [55,60] as well as observational data showing improved outcomes [58], Day et al. performed a phase III RCT comparing induction with AmBd 1 mg/kg/day monotherapy for four weeks (n = 100) against combination therapy with AmBd 1 mg/kg/day + 5-FC 100 mg/kg/day for two weeks (n = 100) and AmBd 1 mg/kg/day + FCZ 800 mg/d for two weeks (n = 99). Treatment with AmBd + 5-FC was associated with a significant reduction in hazard of death at 10 weeks (HR = 0.61, *p* = 0.04) when compared to monotherapy, and, with adjustment for baseline covariates, a significantly reduced hazard ratio compared to both monotherapy and AmBd + FCZ combination therapy at a secondary 6-month time point [61]. While neutropenia was more common in the combination therapy arms, anaemia was seen more frequently with 4-week AmBd monotherapy, and treatment modification or interruption occurred equally across all treatment groups.

### 6.2. Alternative Combinations

These above described trials formed a robust evidence base for the 2010 IDSA cryptococcal guidelines [6] which recommend two weeks of AmBd 0.7–1.0 mg/kg/day + 5-FC 100 mg/kg/day as first line induction therapy for HIV-associated CM, followed by FCZ 400 mg/day for eight weeks before stepping down to FCZ 200 mg/day as consolidation and maintenance therapy, respectively. Despite the efficacy of this induction regimen, there were several barriers to its use in many lower income settings. IV AmBd mandates a prolonged inpatient stay and requires laboratory infrastructure for monitoring renal function, electrolytes, and full blood counts with provision to replace deficits when required. There was also continued lack of access to 5-FC in many low–middle income countries, and where available, 5-FC was often expensive [15,62,63].

### 6.3. Amphotericin B (AmBd) and Fluconazole (FCZ)

FCZ is essentially fungistatic at doses up to 400 mg/day [19]. When used as monotherapy inadequate CSF sterilisation is achieved [19,20] with high associated mortality [64,65]. Based on findings that higher dose FCZ (800 mg) achieves more rapid sterilisation of the CSF [66,67] and that doses of up to 2000 mg appear to be well tolerated [68]. Longley et al. conducted a sequential cohort study comparing FCZ 800 mg/day to FCZ 1200 mg/day, both given as monotherapy. They found an EFA of −0.18 log CFU/mL/day for FCZ 1200 mg/day vs. −0.07 log CFU/mL/day for FCZ 800 mg/day. While this difference was statistically significant, it is well below the EFA of any combination regimen [4,22,28,60,61] and overall 10-week mortality was high at 54%, similar to that seen in other prospective cohort studies of FCZ monotherapy [69].

Higher dose FCZ in combination with AmBd is however a valid alternative therapy where 5-FC is unavailable. Brouwer et al. found the EFA of combination AmBd 0.7 mg/kg/day + FCZ 400 mg to be −0.39 log CFU/mL/day, while not as fungicidal as AmBd + 5-FC combination therapy (−0.54 log CFU/mL/day) it was more fungicidal compared to AmBd alone (−0.31 log CFU/mL/day). Higher doses of FCZ were used in conjunction with AmBd by Pappas et al. in a small phase II RCT designed to compare the safety and tolerability of AmBd 0.7 mg/kg/day monotherapy (n = 47) against AmBb 0.7 mg/kg/day in combination with FCZ at either 400 mg/day (n = 48) or 800 mg/day (n = 45) as a 2-week induction regimen. While underpowered to detect outcome differences, combination therapy was well tolerated and the addition of FCZ 800 mg appeared to result in improved survival at two weeks [70]. Loyse et al. performed a phase II RCT in South Africa comparing four separate regimens of FCZ 800 mg/day, FCZ 1200 mg/day, 5-FC 100 mg/kg/day and voriconazole 600 mg/day in combination with an AmBd backbone [28] and found no significant differences in EFA between all four groups. While subsequent trials powered for mortality endpoints have found that combination AmBd 1.0 mg/kg/day + FCZ 800 mg/day appears less effective than AmBd 1.0 mg/kg/day + 5-FC combination therapy, there was a trend toward lower mortality at ten weeks when compared to an AmBd 1.0 mg/kg/day monotherapy regimen [15,61]

### 6.4. Flucytosine and Fluconazole (5-FC + FCZ)

AmBd requires IV administration, the ability to closely monitor for associated toxicity and is often costly or unavailable in resource poor settings [8,15]; however, the combination of 5-FC with FCZ, has the potential to be a well-tolerated oral regimen which could potentially facilitate earlier hospital discharge. The majority of in vitro studies of triazole–flucytosine interactions report synergy or indifference, with a trend toward increased survival in animal studies over FCZ alone [39]. An early open label trial in which HIV positive patients were treated with a combination of FCZ 400 mg + 5-FC 150 mg/kg/day for ten weeks reported a clinical success rate of 63%, comparable to AmBd + 5-FC combinations at the time, with 95% of participants tolerating therapy for at least two weeks [71] Although of note, these studies predated FCZ maintenance therapy and treatment toxicity was significant in longer AmBd regimens [43,44].

Mayanja-Kizza similarly showed that the addition of 5-FC 150 mg/kg/day to FCZ 200 mg/day in the first two weeks of induction therapy resulted in improved 6-month survival (32% vs. 12% for FCZ 200 mg/day alone, *p* = 0.022) [23]. Milefchik et al. also showed an incremental improvement in the composite outcome measure of survival + CSF clearance at ten weeks as the dose of FCZ was increased from 800 mg to a maximum of 2000 mg. They also found that the addition of 5-FC 100 mg/kg/day to FCZ was independently associated with improved patient outcomes [68]. In an EFA study in Malawi, Nussbaum et al. found FCZ 1200 mg/d with 5-FC 100 mg/kg/day to be more fungicidal than FCZ alone (EFA −0.28 log CFU/mL/day vs. −0.11 log CFU/mL/day, respectively). Even though this was a small phase II study, survival was also significantly better with the combination regimen [22].

### 6.5. Shorter Course Amphotericin B

EFA studies had demonstrated the rapid fungicidal activity of AmBd 1 mg/kg/day, with substantial reduction in fungal burden seen after only one week of treatment [19]. Beyond one week it is arguable that the toxicity associated with AmBd, usually apparent in the second week of therapy [55] may outweigh the continued fungicidal effects [8]. AmBd exhibits concentration-dependent fungal killing with a prolonged tissue half-life, making shorter courses using higher doses an attractive compromise between efficacy and toxicity while also reducing monitoring costs and potentially the length of hospital stay. In Malawi, in a second phase of the study of Nussbaum [22], Jackson et al. added 7 days AmBd 1 mg/kg/day to each of the 14-day treatment arms studied by Nussbaum (FCZ 1200 mg/day monotherapy vs. FCZ 1200 mg/day + 5-FC 100 mg/kg/day). In an overall analysis of both phases, addition of one-week 1 mg/kg/day AmBd and of 5-FC were each independently associated with greater fungicidal activity. Additionally, of note, with FCZ dosed at 1200 mg/day, the triple combination had the most rapid EFA (−0.5 log CFU/mL/day), consistent with the murine studies of Diamond et al. [41]. In Uganda, Muzoora et al. showed that the EFA of five days of AmBd 1.0 mg/Kg/day added to fourteen days of FCZ 1200 mg/day was higher compared to historic controls of FCZ 1200 mg/day monotherapy [72]; moreover, the rate of clearance remained the same in the second week of therapy, even after cessation of AmBd on day five. The authors postulated that sufficiently high tissue doses of AmBd persisted after stopping treatment, and that continued administration of AmBd beyond 5–7 days may not improve fungicidal activity [73], as supported later by data from animal models [74].

Data has previously shown, however, that continuing AmBd into the second week contributed to drug related toxicity. Muzoora found that five days of AmBd was well tolerated and resulted in significantly lower incidence of anaemia when compared to historical 14-day courses of AmBd 1.0 mg/kg/day. A large AmBd toxicity analysis comprising a combined cohort of patients treated with AmBd-based regimens in Africa and Thailand (n = 368) found that grade III/IV anaemia and nephrotoxicity were both significant risk factors for 10-week mortality [7]. The authors concluded that while the electrolyte disturbances and nephrotoxicity of 2-week AmBd regimens can be mitigated with pre-hydration and electrolyte replacement, anaemia is a more concerning adverse effect, with roughly a third of all patients developing a haemoglobin nadir of <7.5 g/dL. Short course AmBd therapy mitigated both nephrotoxicity and anaemia, with significantly higher haemoglobin nadirs (9.5 g/dL vs. 8.3 g/dL, *p* = 0.033) reported in those receiving 5–7- vs. 14-day courses of AmBd combination therapy.

### 6.6. Improved Regimens for Resource Poor Settings

Based on the promising phase II data with oral 5-FC + FCZ and with 1-week AmB-based regimens, Molloy et al. performed a landmark RCT in HIV-CM, enrolling 721 patients in four African countries 1. The aim of the ACTA (Advancing Cryptococcal Treatment for Africa) trial was to identify an induction regimen that was more efficacious than FCZ monotherapy, and safer and easier to implement than two weeks of AmBd 0.7–1.0 mg/kg/day + 5-FC 100 mg/kg/day. Two novel strategies, an all-oral treatment regimen of 5-FC 100 mg/kg/day + FCZ 1200 mg/day for two weeks (n = 225), and AmBd 1 mg/kg/day for one week, were compared with the standard of AmBd 1 mg/kg/day for two weeks. Within the AmBd strategies, participants were randomised 1:1 to either FCZ 1200 mg/day or 5-FC 100 mg/kg/day, as the partner drug, thus creating five unique treatment arms. Mortality was non-inferior, comparing the oral, and one-week AmBd groups with the two-week AmBd group (35.1%, 36.2%, and 39.7%, respectively, at ten weeks). One week of AmBd + 5-FC had the lowest mortality of any regimen, reaching significance compared to the prior gold standard of two weeks’ AmBd + 5-FC, with a HR of death of 0.56 at ten weeks (95% CI, 0.35 to 0.91). The explanation for this difference likely lies in the balance between efficacy and toxicity, with transfusion required in 5.5%, 10.3%, and 22.2% of patients in the all-oral, 1-week AmBd and 2-week AmBd groups, respectively. Nephrotoxicity was also reduced in the all-oral and short-course AmBd arms. Grade 4 neutropenia occurred in 3.2% of patients who received two weeks of 5-FC compared to 0.9% and 1.3% in patients who received 1-week 5-FC, and 5-FC-free regimens, respectively [16].

These findings led the World Health Organisation (WHO) and the Southern African HIV Clinician’s Society to update induction treatment guidelines for HIV-CM to 1 week of AmBd 1.0 mg/kg/day + 5-FC 100 mg/kg/day followed by one week of FCZ 1200 mg/day, with the oral combination regimen recommended where AmB is either not available or is unable to be administered safely [7,75].

Considerable progress has been made with implementing these guidelines, aided by supportive cost-effectiveness analyses [12,76,77]. In South Africa, 5-FC has been used in routine care across the country. In on-going surveillance led by Govender and colleagues, in-hospital mortality for 598 patients given 5-FC-containing regimens (predominantly 1-week AmBd + 5-FC) was 24% compared with 37% for 943 patients on other treatments (predominantly 2-weeks AmB + FCZ), mirroring the benefit of the 1-week regimen seen in the ACTA trial [78]. Outside South Africa, the Unitaid programme, has provided generic 5-FC and treatment support to seven high burden sub-Saharan African countries [17], and early results from an EDCTP-funded implementation project (DREAMM) in Tanzania, Malawi, and Cameroon show similar reductions in mortality as those observed in ACTA and South Africa [79].

### 6.7. Recent Developments and Future Directions

In well-resourced settings with lower incidence of CM, liposomal AmB (L-AmB) is usually used in place of AmBd, with a dose of 3–4 mg/kg/day advised in the 2010 IDSA guidelines [6]. However, there is a relative paucity of robust controlled data regarding its efficacy. One moderately sized randomised trial suggested equivalence to AmBd 0.7 mg/kg/day [80,81] when used as monotherapy, using mycological clearance and mortality as endpoints, but none had examined the efficacy of L-AmB in combination with azoles or 5-FC. L-AmB is also significantly less nephrotoxic than AmBd [80,81], particularly important for immunocompromised patient groups (e.g., solid organ transplant, haematological malignancy with renal impairment, or on concomitant nephrotoxic agents) [82]. Unfortunately, despite being off patent since 2016, L-AmB is difficult to manufacture, and cost has precluded its widespread use up to now in settings where CM incidence is highest [15]. The extremely long half-life and tissue persistence of up to several weeks makes L-AmB preparations perfectly suited as a candidate for high dose short course induction, even as a single dose [11].

A single dose L-AmB treatment strategy has the potential to achieve the rapid fungicidal activity of AmBd-based regimens while enabling earlier hospital discharge and reducing treatment related toxicity. This concept was first trialled and found to be effective in the field of visceral leishmaniasis [83]. For HIV-associated CM, Jarvis et al. compared single dose Ambisome (10 mg/kg) on day 1, two doses of Ambisome with a second 5 mg/kg dose given on day 3, or three doses of Ambisome on days 1, 3, and 7, against the usual daily dosing at 3 mg/kg/day, all on the background of FCZ 1200 mg/day, in a phase II EFA study in Botswana and Tanzania. The abbreviated schedules were all non-inferior in EFA to daily dosing, with no suggestion of additional effect with additional doses [84].

The single high dose strategy was taken forward into a multicentre non-inferiority, phase III trial, AmBition-CM, comparing a single dose of 10 mg/kg of L-AmB on an optimised oral backbone of 14 days of FCZ 1200 mg/day + 5-FC 100 mg/kg/day against the new WHO recommended induction regimen of AmBd 1.0 mg/Kg/day + 5-FC 100 mg/kg/day for seven days followed by FCZ 1200 mg/d for seven days (based on ACTA trial results). 10-week mortality rate in the L-AmB arm was 24.8% (95% CI; 20.7 to 29.3) vs. 28.7% (95% CI; 24.4 to 33.4) in the control, giving a risk difference of −3.93% (90% CI; −9.0 to 1.2), easily meeting the prespecified 10% non-inferiority margin. Anaemia, nephrotoxicity and thrombophlebitis all occurred less frequently in the L-AmB group and both regimens had comparable EFAs [85]. The Ambition-CM trial results confirmed the efficacy of the ACTA trial regimen of one-week AmB + 5-FC, with mortality under 30%, considerably lower than prior results with two-week AmB regimens in Africa [86]; however, the single dose L-AmB regimen offers a practical, easier-to-administer, and even better-tolerated treatment option for HIV-associated CM in low resource settings. A cost effectiveness analysis built into the Ambition-CM trial is underway, but savings from reduced monitoring requirements, and the possibility of earlier discharge in the least sick patients, may help to balance the extra drug cost of L-AmB.

These recent trial findings herald an exciting new era in combination treatment of HIV-associated CM in settings where the burden is highest and resources limited. What is now required is a continued commitment to improve access to and affordability of flucytosine, conventional AmBd and L-AmB, and support for implementation efforts. Gilead Sciences have committed to provide L-AmB for CM treatment at the preferential price (USD 16.25 for 50 mg) as agreed for treatment of leishmaniasis [87]. The Unitaid programme is now providing L-AmB in addition to 5-FC, and some countries have already successfully applied for continued 5-FC supply through the Global Fund [88]. Sustained advocacy and efforts from national HIV programmes and major non-governmental organisations (NGOs) with support from pharmaceutical companies could make improved treatment outcomes accessible to all [15,87,89]. Indeed, the CDC Mycology Branch with other stakeholders, have recently launched an ambitious target to end HIV-associated cryptococcal deaths by 2030 [90].

## 7. Conclusions

There have been significant advances in the treatment of CM through optimisation of combination antifungal therapies, as summarised in Table 1, with phase III trial design based on prioritisation of combinations, dosages, and schedules of existing drugs using randomised phase II EFA trials, and support from animal PK/PD modelling.

HIV-CM treatment is one of the best-evidenced amongst invasive fungal infection, indeed in infectious diseases as a whole, where it serves as a model for the benefit of combination therapy. Hand-in-hand with developments in and rollout of antiretroviral therapy, clinical trials in HIV-infected patients over the decades have demonstrated improving success rates in terms of mortality and mycological clearance using currently available drugs; however, access and cost remain a barrier, particularly in low- and middle-income settings where demand is greatest. The development of surrogate markers of treatment efficacy were key to the success of developing novel combinations regimens as well as optimising dosing, markers that are now being deployed in the evaluation of novel agents in the antifungal armamentarium against Cryptococcus, including APX2039 and oral (encochleated) amphotericin [91]. This successful framework used to identify better CM regimens provides a potential roadmap for other mycoses with high associated mortality despite current monotherapies, and clinical outcomes threatened by increasing drug resistance.

## Figures and Tables

**Table 1 jof-07-01098-t001:** Key randomised controlled CM treatment trials with primary clinical endpoints in various settings over 40 decades.

Year	Author/Setting	Design	Number Participants	Treatments Compared	Primary Outcome	Results	CSF Clearance
1979	Bennett [43] * USA	Open-label, multi-centre (10), non-inferiority (20% margin)	66 evaluable (32 vs. 34) (Stopped early) Non-HIV	10 weeks AmBd monotherapy (0.4 mg/kg/day) vs. 6 weeks AmBd (0.4 mg/kg/day) with 5-FC (150 mg/kg/day)	Clinical improvement or cure at 10 weeks	47% (10-week monotherapy) vs. 68% (6-week combination) (*p* > 0.05)	Faster time to CSF culture negativity in combination arm, figures not reported *p* < 0.001
1987	Dismukes [44] USA	Open-label, multi-centre (17), non-inferiority (15% margin)	91 evaluable (2 = HIV)	4 weeks AmBd (0.3 mg/kg/day) + 5-FC (150 mg/kg/day) vs. 6 weeks AmBd (0.3 mg/kg/day) + 5-FC (150 mg/kg/day)	Disease relapse within one year	Relapse rates: 4-week regimen = 27% 6-week regimen = 16% (difference = 11% (90% C.I. −4 to 25) non-inferiority not met	
1990	Larsen [49] * USA	Open-label, single centre, USA	21 evaluable HIV	10 weeks FCZ 400 mg/day vs. 10 weeks AmBd (1-weeks daily 0.7 mg/kg/day + 9 weeks 0.7 mg/kg 3 days/week) + 5-FC 150 mg/kg/day.	Clinical success: patients alive with negative CSF at 10 weeks	FCZ mono therapy, 57% failed vs. AmBd + 5-FC, 0% failed (*p* = 0.04)	Mean time of CSF positive: FCZ 40.6 days vs. Combination 15.6 days
1992	Powderly [51] USA	Open-label, multi-centre (45), non-inferiority (15% margin)	189 evaluable HIV	12 months FCZ 200 mg/day vs. 12 months AmBd 1.0 mg/kg/week	Disease relapse within one year	Relapse rates: 2% FCZ vs. 18% AmBd (*p* = <0.001)	
1997	Van der horst [50] USA	Open-label, 2-part, double-blind, multicentre trial	Step 1: 381 (202 vs. 179) Step 2: 306 (155 vs. 151) HIV	Step 1: 2 weeks AmBd (0.7 mg/kg/day) + 5-FC (100 mg/kg/day) vs. 2 weeks AmBd (0.7 mg/kg/day) alone Step 2 (for those stable or improved): Induction above followed by 8 weeks Itraconazole (400 mg/day) vs. 8 weeks FCZ (400 mg/day)	2 primary outcomes: Clinical: 2 weeks: stable or improved symptoms 10 weeks: asymptomatic Mycological: 2 weeks: CSF culture negative 10 weeks: CSF culture negative	Clinical endpoint: Step one: Combination 78% vs. AmBd monotherapy 83% (*p* = 0.18) Step two: 68% FCZ vs. 70% Itraconazole (no significant mortality difference between groups).	Mycological endpoint met Step one: 70% Combination vs. 51% AmBd alone (*p* = 0.06) Step two: 72% FCZ vs. 60% itraconazole
1997	Leenders [80] * The Netherlands, Australia	Open-label, multicentre (2), Superiority	28 (13 vs. 15)	3 weeks AmBd (0.7 mg/kg/day) monotherapy vs. Ambisome (4 mg/kg/day)	Composite outcome of clinical efficacy and CSF sterility at 10 weeks	10-week clinical response: 87% Ambisome vs. 83% AmBd	14-day culture conversion: 66% Ambisome vs. 11% AmBd
1998	Mayanja-Kizza [23] Uganda	Open-label, single-centre, superiority	50 evaluable (25 vs. 25	2 months FCZ (200 mg/day) monotherapy vs. 2 months FCZ (200 mg/day) + 2 weeks 5-FC (150 mg/kg/day)	All-cause mortality 14 days and 6 months	14-day mortality: 40% FCZ monotherapy vs. 16% FCZ + 5-FC 6-month mortality: 12% FCZ monotherapy vs. 32% FCZ + 5-FC	
2009	Pappas [70] * USA, Thailand	Open-label, multi-centre (8), non-inferiority (10% margin)	139 (47 vs. 48 vs. 45) HIV	2 weeks AmBd (0.7 mg/kg/day) monotherpy vs. 2 weeks AmBd (0.7 mg/kg/day) + FCZ (400 mg/day) vs. 2 weeks AmBd (0.7 mg/kg/day) + FCZ (800 mg/day)	Composite outcome of survival, neurologic stability and negative CSF after 14 days treatment	Day 14 successful outcome: 41% (AmBd monotherapy) vs. 27% (AmBd low dose FCZ) vs. 54% (AmBd high dose FCZ)	
2013	Day [61] Vietnam	Open-label, single-centre, superiority	299 (100 vs. 100 vs. 99) HIV	4 weeks AmBd (1 mg/kg/day) vs. 2 weeks AmBd (1 mg/kg/day) plus 5-FC (100 mg/kg/day) Vs 2 weeks AmBd (1 mg/kg/day) plus FCZ (400 mg twice daily)	Co-primary outcome: All-cause mortality at 14 and 70 days	AmBd+5-FC vs. AmBd alone: 14-day mortality HR = 0.57 (*p* = 0.08). 70-day mortality HR = 0.61 (*p* = 0.04) AmBd+FCZ vs. AmBd alone: 14-day mortality HR = 0.78 (*p* = 0.42). 70-day mortality HR = 0.71 (*p* = 0.13)	EFA: AmBd alone: −0.31 log CFU/mL/day AmBd + 5-FC −0.42 log CFU/mL/day AmBd + FCZ −0.32 log CFU/mL/day *p* = <0.001 for AmBd + 5-FC vs. both other regimens
2018	Molloy [16] Malawi, Zambia, Tanzania, Cameroon	Open-label, multi-centre (9), non-inferiority (10% margin)	678 (225 vs. 224 vs. 229)	2 weeks FCZ (1200 mg/day) plus 5-FC (100 mg/kg/day) vs. 1 week AmBd (1 mg/kg/day) with either FCZ (1200 mg/day) or 5-FC (100 mg/kg/day) vs. 2 weeks AmBd (1 mg/kg/day) with either FCZ (1200 mg/day) or 5-FC (100 mg/kg/day)	All-cause mortality at 2 weeks	18.2% oral regimen 21.9% 1 week AmBd 21.4% 2 weeks AmBd (RD Oral vs. 2 weeks AmBd= −3.18 (95% CI: −10.5 to 4.2); RD 1 week AmBd vs. 2 weeks AmBd= −0.48 (95% CI: −7.1 to 8.1)) 5-FC vs. FCZ as partner drug: 10-week mortality, 31.1% vs. 45.0%, HR = 0.62 (*p* = 0.002)	EFA Oral 5-FC + FCZ −0.26 log CFU/mL/day 1 w AmBd −0.40 log CFU/mL/day 2 w AmBd −0.42 log CFU/mL/day AmBd + FCZ −0.36 log CFU/mL/day AmBd + 5-FC −0.46 log CFU/mL/day
2021	Jarvis/Lawrence [85] Uganda, Botswana, Zimbabwe, Malawi, South Africa	Open-label, multi-centre (7), non-inferiority (10% margin)	814 (407 vs. 407)	L-AmB (10 mg/kg) Day 1 plus 14 days 5-FC (100 mg/kg/day) and FCZ (1200 mg/day) vs. 7 days AmBd (1 mg/kg/day plus 5-FC (100 mg/kg/day) and FCZ (1200 mg/day) then 7 days FCZ (1200 mg/day)	All-cause mortality at 10 weeks	24.8% (LAmB) vs. 28.7% (Control) (RD 3.93, 90%CI −9.0 to 1.2) Non-inferiority met	Await publication

* Trials underpowered to report on mortality alone—Studies designed to report primarily on fungicidal activity/EFA are not included but are summarised by Jarvis et al. [56].

## Data Availability

Not applicable.

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
