# Peer review of "Combination Therapy for HIV-Associated Cryptococcal Meningitis—A Success Story"

_jof, 2021, doi:10.3390/jof7121098_

Round 1
Reviewer 1 Report
Thank you for inviting me to review JOF-1498203, Hurt et al., Combination Therapy for HIV-Associated Cryptococcal Meningitis—A Success Story.
This is a nice piece summarizing some historical and contemporary aspects of combination antifungal therapy trials in CM.
It would be good to be clearer in Table 1’s title to justify which studies are included and excluded. It currently reads as:
Table 1. Key randomised controlled CM treatment trials focusing on clinical endpoints in various settings over 40 decades. * Trials underpowered to report on mortality alone—Studies
designed to report primarily on fungicidal activity / EFA are not included but are summarised by Jarvis et al. [43].
There are some key randomized trials not included (eg. Leenders AIDS 1997) and some trials mentioned in the text (eg. Brouwer et al., Bicanic et al., Loyse et al., Mayanza-Kizza et al., Jackson et al.) are not here. It doesn’t seem to be sample size based, as Larsen’s 21 person study is here. Also Boulware’s COAT study, Makadzange and Bisson’s ART commencement trials are key RCTs with Boulware’s using combination antifungal therapy. There may be perfectly justifiable reasons why this selection of “key papers” occurred.
I would suggest we insert in the title “combination antifungal therapy” rather than simply “combination therapy” as it seems adjuvant therapies are not the focus on this review. Also, the inclusion of Beardsley at al. in Table 1 suggests that adjuvant therapies may be included, but absent are the IFNg trials, and the more recent failed trials of sertraline, tamoxifen etc..
For interest, it may be nice to expound on why we use combination antifungal therapies in CM, while we do not in invasive aspergillosis, mucormycosis, scedosporiosis etc., where indeed combinations are often frowned upon or considered salvage. Some checkerboard susceptibility data even suggest some combinations as antagonistic.
A tricky component not discussed here is the confounding impact of ICP management as clearly this has major bearing on clinical outcomes (ie. antifungal combination type and dose is not the only determinant in these trials).
Thank you for inviting me to review JOF-1498203, Hurt et al., Combination Therapy for HIV-Associated Cryptococcal Meningitis—A Success Story.
This is a nice piece summarizing some historical and contemporary aspects of combination antifungal therapy trials in CM.
It would be good to be clearer in Table 1’s title to justify which studies are included and excluded. It currently reads as:
Table 1. Key randomised controlled CM treatment trials focusing on clinical endpoints in various settings over 40 decades. * Trials underpowered to report on mortality alone—Studies
designed to report primarily on fungicidal activity / EFA are not included but are summarised by Jarvis et al. [43].
There are some key randomized trials not included (eg. Leenders AIDS 1997) and some trials mentioned in the text (eg. Brouwer et al., Bicanic et al., Loyse et al., Mayanza-Kizza et al., Jackson et al.) are not here. It doesn’t seem to be sample size based, as Larsen’s 21 person study is here. Also Boulware’s COAT study, Makadzange and Bisson’s ART commencement trials are key RCTs with Boulware’s using combination antifungal therapy. There may be perfectly justifiable reasons why this selection of “key papers” occurred.
I would suggest we insert in the title “combination antifungal therapy” rather than simply “combination therapy” as it seems adjuvant therapies are not the focus on this review. Also, the inclusion of Beardsley at al. in Table 1 suggests that adjuvant therapies may be included, but absent are the IFNg trials, and the more recent failed trials of sertraline, tamoxifen etc..
For interest, it may be nice to expound on why we use combination antifungal therapies in CM, while we do not in invasive aspergillosis, mucormycosis, scedosporiosis etc., where indeed combinations are often frowned upon or considered salvage. Some checkerboard susceptibility data even suggest some combinations as antagonistic.
A tricky component not discussed here is the confounding impact of ICP management as clearly this has major bearing on clinical outcomes (ie. antifungal combination type and dose is not the only determinant in these trials).

Reviewer 2 Report
This review summarises the main findings of clinical trials that support the successful of antifungal drug combination. Some minor points in the text should be proofread and edited.
Author Response
Thank you for taking the time to review our article, it is greatly appreciated.
We have proof read the text and changed errors encountered. Including changing the title of the table to “Table 1. Key randomised controlled CM treatment trials with primary clinical endpoints in various settings over 40 decades” to better reflect the articles included.
Reviewer 3 Report
The article by Hurt et al critically reviews the importance of different combination therapy in treating Cryptococcal meningitis (CM). The review article is well written and can be a good source to quickly refer to while studying combination therapy. Here are my minor comments:
1)The reviewer feels that the review article will greatly benefit if it discusses how monotherapy especially fluconazole monotherapy can lead to drug resistance in Cryptococcus clinical isolates. Discussing author's own paper (https://pubmed.ncbi.nlm.nih.gov/30688656/) in the review will greatly increase the significance of the combination therapy. Several studies have discussed issues regarding fluconazole monotherapy. The author should consider writing a paragraph in discussing the pitfall of the monotherapy.
2)All scientific names need to be italicized (lines 30 and 59).
3)Are other forms of azoles ever used in combination with flucytosine or amB to treat CM? If so please clarify. For instance combination of Posaconazole and AMB was found to be effective in the following study (https://www.hindawi.com/journals/bmri/2020/8878158/)
